# CaTran: ultra-light neural network for predicting gene-gene interactions from single-cell data

## Abstract

Part of the difficulty of learning a gene-regulatory network from expression data is related to the fact that edges in such a network represent different interactions with a different effect size. Therefore modeling gene associations requires learning an individual function for each pair of interacting genes. This may greatly inflate the number of parameters in a model and lead to insufficient generalization. In this paper we propose a method for gene regulatory network inference, called CaTran (Causal Transformer), which avoids explicitly learning pairwise relation between genes, which allows it to significantly reduce the size of the model. The key feature of this approach is learning for each gene a low dimensional embedding and then using a self-attention mechanism to estimate its relation to other genes. Our method is applicable for both observational data and data with interventions. For the latter it implements a differentiable gene importance test and forces attention values to be in accordance with it. Because the gene regulatory network in CaTran is learned as a soft adjacency matrix, it allows sampling graphs with arbitrary number of edges based on a set threshold. Comparison of these graphs with the gene networks from databases showed that even for large graphs the edges are predicted with high precision.

## 1 Model description

Here we present our solution to CausalBench challenge (Chevalley et al., 2022).

### 1.1 Data preprocessing

Our experiments have shown that any additional preprocessing of the data at most yields no increase in model performance as compared to running it using raw counts. We have tried different methods of normalization including scanpy standard pipeline (Wolf et al., 2018) and CLR transform (Stoeckius et al., 2017). The decrease in performance is likely to be associated with the spurious correlation patterns which arise in data after normalization. Then we also tried imputing data using various techniques such as MAGIC (Dijk et al., 2018) and SVD based imputation (replacing zero values with inverse SVD transform). The ineffectiveness of applying these methods indicates the importance of zeros in data as a biological signal for predicting gene regulatory networks (Jiang et al., 2022).

### 1.2 Training objective

CaTran is built upon the DCDI framework (Brouillard et al., 2020) simultaneously simplifying it and regularizing its behavior. From its predecessor, CaTran inherits the basic outline of the learning objective. CaTran does not directly optimize inference of gene interactions but instead solves gene expression prediction tasks. And in the end it uses some of the model parameters as a proxy for gene interaction scores. Unlike DCDI, however, CaTran does not encode these scores explicitly as a learnable adjacency matrix but computes them using self-attention mechanism. Another key distinction of CaTran from DCDI is that instead of modeling distribution of gene expression it

directly predicts the expression of genes in a minibatch. We did this because the expression of genes in single-cell data does not follow any parametric probability distribution.

The model is trained using mini-batches which include a subset of cells and a subset of genes. The typical size of a mini-batch is 2048 cells and 500 genes. If the number of genes is less than 500 genes, then the mini-batch includes all genes. Using large batches with more than 1000 genes resulted in decreased performance. In each mini-batch a fraction of genes sampled randomly is perturbed by shuffling values between selected genes. Initially, we tried zeroing out these genes but the new strategy yielded better results. We also experimented with augmenting different fractions of the input and established empirically that hiding expression of as much as 80% genes leads to better results. Overall, this strategy is reminiscent of how the language models such as language transformers are trained (Devlin et al., 2019).

The objective of a neural network is to predict the true values of genes with augmented expression. And so its loss function consists of three terms, two of which correspond to this task. The model separately calculates the predicted expression of augmented genes and genes with unaugmented expression and calculated Huber loss (Gokcesu & Gokcesu, 2021). These two terms combined with different weights, correspondingly 0.7 and 0.3. The second term is added to control that the model will not forget the true values of gene expression of genes with unaugmented expression. The choice of Huber loss rather than MSE is crucial to maintain high performance of the program because it reduces the effect of outliers.

To optimize the given objective we used Lion optimizer(Yazdani & Jolai, 2016). We compared it to AdamW(Loshchilov & Hutter, 2019) and found it more preferable. By default we use it for 25 epochs with low learning rate (0.001) and weight decay (0.05). The model weights are initialized with values from normal distribution with zero mean and the standard deviation of 0.001. The choice of this initialization strategy was dictated by the fact that we used SiLU as an activation function (Elfwing et al., 2017).

## 1.3 CaTran architecture

The guiding principle for implementing CaTran architecture (Figure 1A) was the idea that interactions between genes can be encoded in learnable gene embeddings. This helps to avoid learning these interactions explicitly. In contrast the original DCDI approach implements learning the whole adjacency matrix which is quadratic to the number of genes. Similarly, CellOracle trains a separate linear model for each network edge (Kamimoto et al., 2023). In our model we compress this information in low dimensional embeddings. The manual search indicated that the optimal embedding size is 40. Though it is a very robust hyperparameter and its alterations do not affect the performance of the program dramatically. Using embeddings also allows to reduce the number of genes used in a mini-batch.

CaTran next uses these embeddings to estimate interactions between genes using self-attention. We tried its different implementations and in the end came up with the following structure. The embeddings are passed to a linear layer which uses the same weights to transform each embedding, then the matrix of dot products between these embeddings is estimated. Then we apply softmax to this matrix along the dimension which conceptually represents the incoming edges in a gene regulatory network. The resulting scores approximate gene interaction scores. We tried binarizing this matrix based on a selected threshold as proposed by DCDI but it led to a drop in performance. This indicates that gene-gene connectivity on its own is not enough to model gene interactions.

After attention weights have been estimated, the model modifies embeddings by adding to them gene expression values. They are then passed through two linear layers with batch normalization layers between them but without activation. This empirically led to better results, which can be related to the issues with numerical instability, since the embeddings are initialized with very small values. Then modified embeddings are passed to an attention block (Figure 1B), which updates each gene embedding based on embeddings of other genes using the precomputed attention weights. They are then passed to batch normalization layer, linear layer, another batch normalization layer and in the end to a non-linear activation function. Finally, the output of the attention block is passed to two linear layers which produce the output.

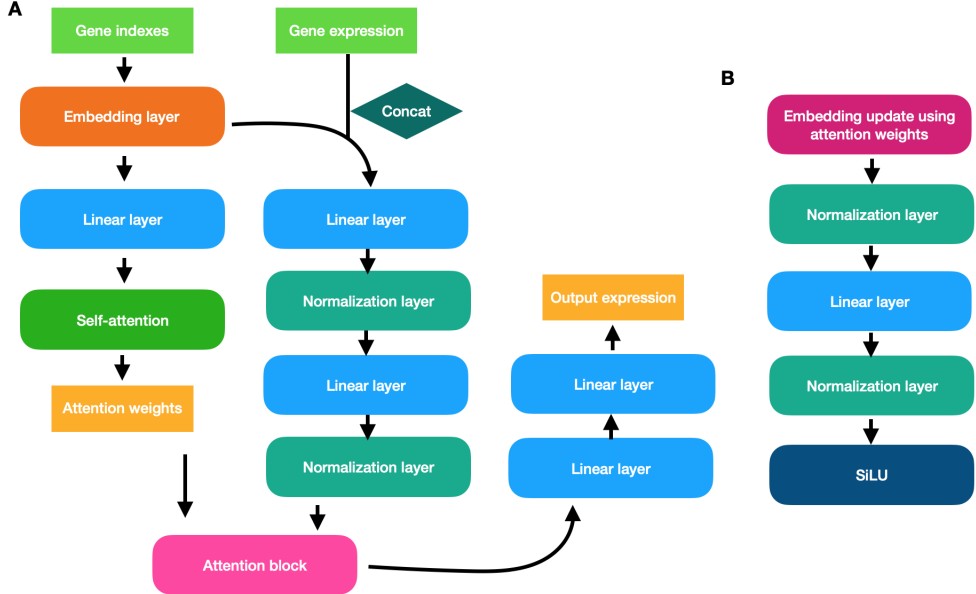

Figure 1: Figure 1. Schematic of CaTran.
**A** The basic outline of the model. **B** Schematic of the attention block.

## 1.4 INTERVENTIONAL LOSS

Though CaTran is able to make accurate predictions in observational regime, its true power is achieved when used with interventional data. To make use of the knowledge about perturbed genes, we introduce interventional loss term into our model. Its purpose is to make attention weights follow importance scores calculated based on the analysis of associations between a perturbed gene and all other genes. The idea behind these scores is inspired by Wu et al., (2023). In essence its premise is that if one gene is associated with another, then its expression should help the model to make accurate predictions. And so for each perturbed gene in a mini-batch we estimate an error of gene expression predictions within cells where this gene is active and within cells where it was turned off, then we take a ratio of these two error estimates, subtract one and transform using sigmoid:

$$Sigmoid(\frac{Huber(non\_interv\_pred, non\_interv\_true)}{Huber(interv\_pred, interv\_true)/)} - 1)$$

. Then we penalize attention coefficients using Huber loss if they deviate from the importance scores.

## 1.5 RETRIEVING AN ADJACENCY MATRIX

In the end of the training CaTran learns embeddings, which can be used to predict an association between genes. In order to get an adjacency matrix, we calculate pairwise dot products between embeddings and then transform them using softmax. Finally, CaTran ranks all edges in this soft adjacency matrix by their attention weight and sets 1 to top 1000 genes and 0 to the rest. One can also easily sample graphs with an arbitrary number of edges.

CaTran outputs directed graphs, however unlike DCDI it allows the existence of cycles in the graph. It was done intentionally, since as we noticed the biological networks do not conform to acyclicity constraints.

## 1.6 IMPLEMENTATION DETAILS

The model is implemented using PyTorch and PyTorch Lightning. As it follows from the title of this paper our model uses comparatively few learnable parameters. The total number of parameters can be estimated using the following formula: $9000 + 40 * number\_of\_genes$.

## 2 CITATIONS

Brouillard, P., Lachapelle, S., Lacoste, A., Lacoste-Julien, S., & Drouin, A. (2020). Differentiable Causal Discovery from Interventional Data (arXiv:2007.01754). arXiv. https://doi.org/10.48550/arXiv.2007.01754

Chevalley, M., Roohani, Y., Mehrjou, A., Leskovec, J., & Schwab, P. (2022). Causal-Bench: A Large-scale Benchmark for Network Inference from Single-cell Perturbation Data (arXiv:2210.17283). arXiv. https://doi.org/10.48550/arXiv.2210.17283

Devlin, J., Chang, M.-W., Lee, K., & Toutanova, K. (2019). BERT: Pre-training of Deep Bidirectional Transformers for Language Understanding (arXiv:1810.04805). arXiv. https://doi.org/10.48550/arXiv.1810.04805

Dijk, D. van, Sharma, R., Nainys, J., Yim, K., Kathail, P., Carr, A. J., Burdziak, C., Moon, K. R., Chaffer, C. L., Pattabiraman, D., Bierie, B., Mazutis, L., Wolf, G., Krishnaswamy, S., & Pe'er, D. (2018). Recovering Gene Interactions from Single-Cell Data Using Data Diffusion. Cell, 174(3), 716-729.e27. https://doi.org/10.1016/j.cell.2018.05.061

Elfwing, S., Uchibe, E., & Doya, K. (2017). Sigmoid-Weighted Linear Units for Neural Network Function Approximation in Reinforcement Learning (arXiv:1702.03118). arXiv. https://doi.org/10.48550/arXiv.1702.03118

Gokcesu, K., & Gokcesu, H. (2021). Generalized Huber Loss for Robust Learning and its Efficient Minimization for a Robust Statistics (arXiv:2108.12627). arXiv. http://arxiv.org/abs/2108.12627

Jiang, R., Sun, T., Song, D., & Li, J. J. (2022). Statistics or biology: The zero-inflation controversy about scRNA-seq data. Genome Biology, 23(1), 31. https://doi.org/10.1186/s13059-022-02601-5

Kamimoto, K., Stringa, B., Hoffmann, C. M., Jindal, K., Solnica-Krezel, L., & Morris, S. A. (2023). Dissecting cell identity via network inference and in silico gene perturbation. Nature, 614(7949), Article 7949. https://doi.org/10.1038/s41586-022-05688-9

Loshchilov, I., & Hutter, F. (2019). Decoupled Weight Decay Regularization (arXiv:1711.05101). arXiv. https://doi.org/10.48550/arXiv.1711.05101

Stoeckius, M., Hafemeister, C., Stephenson, W., Houck-Loomis, B., Chattopadhyay, P. K., Swerdlow, H., Satija, R., & Smibert, P. (2017). Simultaneous epitope and transcriptome measurement in single cells. Nature Methods, 14(9), Article 9. https://doi.org/10.1038/nmeth.4380

Wolf, F. A., Angerer, P., & Theis, F. J. (2018). SCANPY: Large-scale single-cell gene expression data analysis. Genome Biology, 19(1), 15. https://doi.org/10.1186/s13059-017-1382-0

Wu, A. P., Markovich, T., Berger, B., Hammerla, N., & Singh, R. (2023). Causally-guided Regularization of Graph Attention Improves Generalizability (arXiv:2210.10946). arXiv. http://arxiv.org/abs/2210.10946

Yazdani, M., & Jolai, F. (2016). Lion Optimization Algorithm (LOA): A nature-inspired metaheuristic algorithm. Journal of Computational Design and Engineering, 3(1), 24–36. https://doi.org/10.1016/j.jcde.2015.06.003

