# OpenReview forum: "CaTran: ultra-light neural network for predicting gene-gene interactions from single-cell data"
_GSK.ai/2023/CBC_

### Official Review · Reviewer_QXc2 · 2023-04-27
**Interesting method explained in a well-written report**

**Rating:** 7
**Confidence:** 4

**Review:**

Reviewer's summary:

This work proposes CaTran (Causal Transformer) which learns a low-dimensional embedding for each gene and uses a self-attention mechanism to estimate its relation to other genes. Leveraging the low-dimensional embedding, the model requires fewer parameters and is argued to generalize better. The method is built upon DCDI, but instead of directly predicting the interaction network, it predicts the gene expression and the interactions are learned alongside as a side-product of the method. It also generalizes the distributional assumption of DCDI by direct sampling from the distribution instead of putting parametric distributional constraints on the values of the gene expressions.\
&nbsp;

Reviewer's comments:

In contrast to many existing works that explicitly model the interaction network, this work stands out by making the assumption that a soft interaction network can be learned through embeddings for each gene that encodes its interaction with other genes. In addition to computational benefits, this assumption makes the GRN problem amenable to attention-based methods such as Transformers.
The report is easy to follow and the authors have provided a fair amount of details on the decisions made in designing their algorithms. The report can benefit from pseudocode and more implementation details in longer manuscripts but it is sufficient for a short report.